# Significantly Enhanced Electrical Performances of Eco-Friendly Dielectric Liquids for Harsh Conditions with Fullerene

**DOI:** 10.3390/nano9070989

**Published:** 2019-07-09

**Authors:** Zhengyong Huang, Feipeng Wang, Qiang Wang, Wei Yao, Kai Sun, Ruiqi Zhang, Jianying Zhao, Ziyi Lou, Jian Li

**Affiliations:** 1State Key Laboratory of Power Transmission Equipment and System Security and New Technology, Chongqing University, Chongqing 400044, China; 2Postdoctoral Research Station on Chemical Engineering and Technology, Chongqing University, Chongqing 400040, China; 3State Grid Chongqing Electric Power Company, Chongqing 401123, China

**Keywords:** eco-friendly, dielectric liquids, fullerene, electrical performance

## Abstract

The eco-friendly vegetable liquid is increasingly used because of the growing demand for environmentally friendly dielectric liquid. A vegetable liquid/fullerene nanofluid was fabricated via ultrasonic processing with good dispersion of the fullerene nanoparticles. It was observed that a small amount of fullerene (~100 mg/L) can significantly improve the electrical properties of vegetable insulating liquid (dissipation factor decreased by 20.1%, volume resistivity increased by 23.3%, and Alternating Current (AC) dielectric breakdown strength increased by 8.6%). Meanwhile, the trace amount of fullerene is also able to improve the electrical performances (i.e., dissipation factor and electrical resistivity) of the vegetable nanofluid under harsh conditions of long-term thermal aging compared with the blank contrast. The reduced acid values (25%) and dissolved decomposition gases (58.2% for hydrogen) in the aged vegetable nanofluid indicate the inhibition of molecule decomposition of vegetable liquid with fullerene. The improved electrical performances and thermal resistance of the vegetable nanofluid contribute to the electron affinity of fullerene proved by calculation of electron density distribution on the surface. The thermogravimetric analysis of the nanofluid under different atmospheres interprets that the oxygen absorbed inevitably in the fullerene contributes to the performance deterioration of the nanofluids during the initial aging. This work provides a potential method towards eco-friendly dielectric liquid with great electrical performances for harsh environments.

## 1. Introduction

Liquid dielectric, which is a self-healing dielectric material in a liquid state, is widely used as dielectric materials in capacitors [1,2] and cables [3], and as an insulating coolant in transformers [4] and switchgears [5] for the non-permanent conductive trace in the fluid in the discharge channel. To further reduce the fire risk from the liquid dielectric and improve the reliability and eco-friendliness of the insulating facility, researchers are pursuing eco-friendly liquid dielectric with superior electrical performances from the natural resources, such as natural esters [6,7]. The electrical properties of the eco-friendly liquid dielectric generally include dissipation factors, volume resistivity and dielectric breakdown strength [8]. The electrical properties tend to be strongly influenced by dissolved gases, dust, and especially ionic impurities such as acidic material [9]. Recently, some studies have reported that the incorporation of inorganic nanoparticles into natural esters can greatly improve the electrical performances of liquid dielectric materials [10,11,12,13]. Given the extremely high specific surface area and reactivity, a few nanoparticles can absorb the reactive oxygen species produced during the aging of vegetable insulating liquid and inhibit its oxidation. Meanwhile, nanoparticles adsorbed the moisture reduced the hydrolysis of ester molecules and improved the anti-aging effect of oil–paper insulation [14].

Nevertheless, it is worth noting that the dispersibility of magnetic nanoparticles was greatly influenced by orientation of the external magnetic field [15]. In magnetic fields, the magnetic particles aggregation led to the formation of the bridge across the gap between the electrodes, which lowered the dielectric breakdown voltage and was not propitious for wide use in transformers [16]. There exists the influence of magnetic fields on dielectric properties in the ester nanofluids modified by conductive nanoparticles [17]. Thus, the overall electrical performances of the modified liquid dielectric may not be improved [18,19]. Furthermore, because of the high ratio of inorganic nanoparticle fillers, the biodegradability of the natural esters has been much lower than the fresh natural esters [20,21]. Therefore, the basic advantage of natural esters being renewable disappears. This restricts extensive applications of natural materials in electrical and electronic industries. Thus, an eco-friendly liquid nanodielectric with overall improved electrical performances and lower filler concentration under harsh conditions is highly desired.

Fullerene with unique physico-chemical properties [22] have been widely investigated as multifunctional materials for applications in tissue engineering [23], photovoltaics [24], molecular imaging, and bio-sensing [25,26]. Specifically, it has been proved that the addition of fullerene C_60_ to mineral oil enhances the resistivity by 20–30% and reduces the dielectric loss by one to two orders of magnitude [27,28,29].

In this study, it is demonstrated for the first time that a small amount of fullerene (~100 mg/L) significantly improves all the electrical properties of vegetable insulating liquid (Refinement Bleaching and Distillation (RDB)) under harsh conditions of long-term thermal aging compared with the blank contrast. The reduced acid values (25%) and dissolved decomposition gases (58.2% for hydrogen) in the aged vegetable nanofluid (fullerene RDB nanofluid) indicate the inhibition of molecule decomposition of vegetable liquid with fullerene. The improved electrical performances of the vegetable nanofluid under thermal aging contribute to the electron affinity of fullerene. The thermogravimetric analysis of the nanofluids under different atmospheres indicates that the oxygen absorbed inevitably in the fullerene contributes to the performance deterioration of the nanofluids during initial aging. This work provides a potential method towards eco-friendly dielectric liquid with great electrical performances for harsh environments.

## 2. Experimental

### 2.1. Preparation of C_60_ Nanofluid

Firstly, 0.5 g of oleic acid was taken and a small amount of anhydrous ethanol was mixed with C_60_ and mechanically stirred for 2 h to maintain ultrasonic dispersion. The ultrasound energy was 455 W and the ultrasonic process was a circulation of a 2 s pause and a 2 s operation. Secondly, it was cooled to room temperature and subsequently centrifuged for 3000 rpm and washed with ethanol and cyclohexane several times to remove the residual unreactive oleic acids on the surface of C_60_. Thirdly, the mixed liquor was placed into the vacuum drying oven at 60 °C, and the vacuum degree was maintained at 0.1 MPa for 12 h. Finally, the modified C_60_ nanoparticles were prepared and reserved.

The vegetable liquids (RDB obtained from raw rapeseed oil by three procedures, alkaline refinement, vacuum distillation, and bleaching in Chongqing University [30]) and mineral liquids (25# from Karamay, Xinjiang, China) were dried under a pressure of 0.1 MPa for 72 h at 60 °C. The nanoparticles were added to two insulating liquids at the concentration of 0, 50, 100, 200, and 300 mg/L. The C_60_ nano-modified insulating liquid was ultrasonically agitated for 20 min at 30 °C to uniformly disperse nanoparticles in the liquid. 

### 2.2. Electrical Test

Different concentrations of C_60_ modified insulating liquid also present different performances. In this paper, the properties of vegetable liquid (RDB oil) were measured by an authorized testing institute in China, applying Chinese testing standards in accordance with the International Electro-technical Commission (IEC) and International Organization for Standardization (ISO) [30]. Particularly, the measurement of dielectric loss and electrical resistivity were performed using BAURDLTC dissipation factor measurement equipment, followed by the execution of IEC 60247. The measurement of AC breakdown voltage was performed using flat spark gap at ambient temperatures and AC voltage with a frequency of 50 Hz. A 2.5 mm spark gap was used in the test. The measurement of lightning impulse breakdown voltage followed IEC 60897, and a 15-mm needle-sphere spark gap was used. Each measurement was carried out five times, and the arithmetic mean was obtained.

### 2.3. Thermal Aging Test

According to IEC 61125A, the thermal aging speed was doubled when the aging temperature of the insulation liquid was increased by 6 °C. This study analyzed the nanofluid samples without C_60_ after aging. A total of 28 samples were analyzed using different aging times (0, 6, 12, 18, 24, 30, and 36 days), and various concentrations of C_60_ (0, 50, 100 and 200 mg/L) accelerated aging of the samples at 130 °C. Firstly, the C_60_ nano-modified vegetable insulating liquid was dried under a pressure of −0.1 MPa for 48 h at 90 °C. Secondly, when the sample was cooled to room temperature, the samples were packed into a conical flask, and most of the air in the bottle was discharged by nitrogen. Finally, glass bottle stoppers were used to seal the sample bottles after the injection of nitrogen, and the bottles were wrapped with PMP(Poly(4-methylpentene-1)) polyethylene film and aluminium foil. The PMP polyethylene film is utilized to ensure the isolation of nanofluids with the oxygen in the air during the aging process and to keep the thermal aging of nanofluids under the nitrogen atmosphere. The aluminium foil was chosen to prevent the nanofluids from the light irradiation because of the strong photosensitivity of the C_60_ nanoparticles.

### 2.4. Simulation of Electron Density Distribution of C_60_

The electron density distribution of C_60_ was calculated by software Gaussian 09W (Gaussian Inc, Wallingford, CT, USA), which specializes in chemical analysis. Then, the graphical result of the electron density distribution was shown by using software of Gauss View.

## 3. Results and Discussion

### 3.1. Characterization of Nanofluids

Although C_60_ have minimal solubility in organic solvents, the extremely small particle size of its nanoparticles has a large specific surface area with high surface energy, which leads to agglomeration. Thus, oleic acid was used in this experiment to modify the surface of C_60_ nanoparticles, thereby enhancing dispersibility and stability of C_60_ in insulating liquid. 

Figure 1 shows the infrared spectroscopy pattern (test by Thermo Fisher NICOLET IS10, Shanghai, China) of C_60_ nanoparticles. The characteristic peaks of C_60_ were clearly indicated at 524, 574, 1182, and 1428 cm^−1^ and peaks at 524 and 574 cm^−1^ were most evident, which was assigned to the four main Infrared(IR) bands, dipole-active vibrational modes with F_lu_ symmetry, of C_60_ molecule [31]. The vibration peaks at 2850 and 2919 cm^−1^ were absorbed by saturated hydrocarbon (–CH_3_ and –CH_2_) of oleic acid molecules. Moreover, the peak of carboxylate, which was in the range of 1540 cm^−1^, indicated the presence of oleic acid around the nanoparticles, and the peak of CO_2_ was also evident [32].

Figure 2 shows the X-ray diffraction (XRD test by Shimazu XRD-6000, Cu-Kα) of C_60_ and the peak effect of its size and shape. The size of the nanoparticles was calculated using the Scherrer Equation [33]:(1)$$D=\frac{K\cdot \lambda }{\beta \cdot cos\theta }$$
where *D* is the size of a crystal particle, *β* is the half-width of diffraction peak, *θ* is the diffraction angle of X-ray, and *K* is constant with a value of 0.89. The result showed the size of C_60_ crystal particle at a point with 4 nm to 6 nm. Table 1 shows the basic physical and chemical properties of pure vegetable insulating liquid (RDB fabricated by Chongqing University [30]) and mineral liquid (25# from Karamay, Xinjiang). The nanofluids are expected to be processed away from light due to the photosensitivity of C_60_. The double bond of the carbon in C_60_ molecule can be opened in certain light conditions, and then adjacent C_60_ molecules might be linked by new covalent bonds [34]. This characteristic may cause the unsatisfactory modification and dissolution of C_60_ nanoparticles.

The zeta potential of C_60_ modified insulating liquids was tested. Zeta potential is an important characterization of the stability of a dispersion system. The absolute value of zeta potential of a stable dispersion system requires at least 30 mV. The result showed that the zeta potential of 300 mg/L concentration of C_60_ nanoparticles for vegetable insulating and mineral liquids was −45.7 mV and 39.4 mV, respectively, thereby indicating that the modified C_60_ nanoparticles in insulating liquid were stabilized. The nanofluids with different concentrations of C_60_ (50–300 mg/L) were stable and without sediment even after 12 months.

### 3.2. Dielectric Loss and Electrical Resistivity

The dielectric dissipation factor and electrical resistivity can effectively reflect the degradation and contamination of insulating liquids. Figure 3 shows that the dissipation factor and electrical resistivity of nanofluids changed with different concentrations of C_60_, and the measured results of vegetable liquid varied a little greater than mineral liquid. For vegetable liquid, the greatest variation in the dissipation factor and electrical resistivity occurred at 0 mg/L and 50 mg/L, respectively. As for mineral liquid, the greatest variation in the dissipation factor and electrical resistivity occurred at 250 mg/L and 50 mg/L, respectively. Figure 3a shows that the dissipation factor of the vegetable insulating liquid significantly decreased at a low concentration from 50 mg/L to 150 mg/L, and then returned to the situation of the pure sample. The largest drop appeared in 100 mg/L concentration of C_60_, which shows an approximate decline of 20.1%. The experimental data evidently showed that the electrical resistivity of nanofluid increased after C_60_ adjunction. The overall trend exhibited a decrease after the first increase, followed by the increasing concentration of C_60_. The electrical resistivity of vegetable insulating liquid obtained the maximum upgrade at the concentration of 100 mg/L, which showed an approximate increase of 23.3%. Although the concentration reached up to 300 mg/L, electrical resistivity also increased to nearly 1.3 × 10^10^ Ω·m.

Figure 3b shows that the addition of C_60_ nanoparticles would not have a significant effect, except at 50 mg/L. This result was evident due to the low dissipation factor resistivity of mineral liquid. However, the electrical resistivity of mineral based nanofluid decreased significantly when the concentration was greater than 50 mg/L. Thereby, addition of C_60_ nanoparticles in liquid can improve the dielectric properties for vegetable insulating liquid, while it does not change obviously for mineral liquid.

### 3.3. Dielectric Breakdown Strength

AC breakdown voltage is an important parameter that characterizes the dielectric strength of the liquid medium. Figure 4 shows that the AC breakdown voltage of nanofluid varied with different C_60_ concentrations, and the measurement of vegetable liquid varied a little greater than mineral liquid, consistent to electrical resistivity and the dissipation factor. The greatest variation of AC breakdown voltage of vegetable liquid occurred at 0 mg/L, and that of mineral liquid occurred at 150 mg/L. The breakdown voltage may not be negatively affected due to the doped C_60_ into the vegetable insulating liquid. The AC breakdown voltage slightly increased at low concentrations and then significantly declined. The AC breakdown voltage obtained the most improvement with an increase of approximately 8.6% at 100 mg/L concentration of C_60_. The breakdown voltage decreased by 11.3% compared to pure liquid following the increasing concentration. However, mineral liquid obtained the most improvement at approximately 21.7% at 200 mg/L concentration of C_60_. The result indicates that modified vegetable insulating liquid can obtain the optimal dielectric properties and the AC breakdown characteristic at 100 mg/L concentration of C_60_.

Table 2 shows the lightning impulse breakdown voltage of nanofluids with 100 mg/L C_60_ nanoparticles. The lightning impulse breakdown voltage was enhanced to a certain extent. Given the addition of C_60_ nanoparticles, positive lightning breakdown voltage increased by approximately 7.3%, and the percentage of negative lightning breakdown voltage was 7.4% greater than vegetable insulating liquid. As a control for mineral liquid, the lightning impulse breakdown voltage of the mineral insulating liquid was simultaneously promoted through the modification of C_60_ nanoparticles. The positive and negative breakdown voltages increased by 10.0% and 7.6%, respectively.

The breakdown characteristics of modified insulating liquid with C_60_ under AC and impulse voltage can be unified into the streamer development [35,36]. In the development of the streamer of nanofluids, electron mobility is much faster than the positive ion mobility. Nanoparticles can adsorb the fast electrons and convert them into slower negative charges, which results in reducing the development of the head development rate of the streamer. Thus, this weakens the electric field strength of the head and meanwhile reduces the rate of positive and negative charge movement. Finally, the breakdown voltage and the breakdown time increased.

The time of the electron is captured by nanoparticles in the transformer liquid calculated as [36]: (2)$$\tau =\frac{2\varepsilon_{bf}+\varepsilon_{np}}{2\sigma_{nf}+\sigma_{np}}$$
where *ε_bf_* and *ε_np_* are relative dielectric constant of insulating liquids and nanoparticles, respectively. *σ_nf_* and *σ_np_* are the electrical conductivity of nanofluids and nanoparticles, respectively. Due to the time of the streamer development in insulating liquids is in microseconds, it can be considered that when the relaxation time of the nanoparticle is far less than the microsecond level, electrons can be trapped during the development of the streamer, thereby inhibiting the streamer development [37,38].

As the semiconductor material of C_60_ nanoparticles, the charge distribution is generated on its surface under an electric field. The presence of a surface charge could result in the spatial potential to occur with redistribution around the center of nanoparticles. The model of potentials generated by the spherical charges of nanoparticles in Figure 5 can be expressed by:(3)$$\delta =\varepsilon_{0}E_{0}\left(\frac{\varepsilon_{np}-\varepsilon_{bf}}{2\varepsilon_{bf}+\varepsilon_{np}}\right)cos\varphi sin\theta$$

When the direction of the external electric field is the same as the positive direction of the X-axis, the potential distribution generated by the surface polarization of the nanoparticle is:(4)$$V\left(r,\varphi \right)=\frac{aE_{0}}{4\pi }\left(\frac{\varepsilon_{np}-\varepsilon_{bf}}{2\varepsilon_{bf}+\varepsilon_{np}}\right)\times \int_{-\pi }^{\pi }\int_{0}^{\pi }\frac{{sin}^{2}\theta d\theta cos\varphi d\varphi }{\sqrt{1+{\left(r/a\right)}^{2}-2\left(r/a\right)sin\theta cos\left(\varphi -\varphi^{\prime }\right)}}$$
where the $E_{0}$ is the field strength, r is the distance from the centre of the nanoparticles and the a is radius of the nanoparticles. The results of the nano-polarization model show that the distribution of the surface potential of nanoparticles is also related to their size. Surfactants on surface of nanoparticles, which increased the effective radius of nanoparticles, were increasing the trap depth of the nanoparticles [18]. Subsequently, the potential well was prompted to deepen and increased the breakdown characteristics. However, as soon as the concentration of C_60_ nanoparticles increased to a certain extent, the percolation mechanism occurred; that is, the nanoparticles form the semi-conductive parts where C_60_ is the conductor and then reduce the breakdown strengths.

However, C_60_ nanoparticles also have their unique side, such as electronegativity. The C_60_ molecule contains 60 electrons, but its closed shell structure requires 72 electrons. Theoretical calculations show that the lowest unoccupied molecular orbital (LUMO) energy level of the C_60_ molecule is low and is in triple degeneration, which allows a single C_60_ molecule to accept at least six electrons, thereby leading to strong electronegativity [35]. Furthermore, electron affinity, which reflects the energy released by a unit atom or molecule that captures an electron, can respond to the capacity of the atom or molecule to accept electrons. The greater the affinity of the electron, the stronger is the capability of atoms or molecules to capture electrons. On the contrary, the electrons are more likely to escape. Moreover, if the electron affinity value is equal to or even less than zero, then the surface charge escapes at any time. At present, the electron affinity of C_60_ was accurately calculated by Wang et al. (2.683 ± 0.008 eV) [37]. C_60_ molecules capture free electrons to form negative ions, thereby weakening the discharge development and enhancing the breakdown strengths of vegetable insulating liquid.

### 3.4. Dielectric Properties of Aging Nanofluid

Figure 6 shows that the dissipation factor changed with the aging time of vegetable and mineral liquids with different concentrations of C_60_. The increase rate of dielectric loss factor of C_60_ modified vegetable liquid was evidently higher than pure liquid in the early stage. After 12 days of aging, the dissipation factor of the modified vegetable liquid slowly changed and the dielectric loss factor became lower than pure liquid. The dissipation factor of pure mineral liquid did not result in any significant change. However, with the addition of C_60_ nanoparticles in mineral liquid, the dissipation factor increased with the concentration.

The C_60_ nanoparticles are able to slow down the thermal aging of the vegetable insulating liquid. However, there exist small amounts of oxygen in the C_60_ nanoparticles inevitably during the experiment. The traces of oxygen introduced into the vegetable insulating liquid by C_60_ can accelerate the cracking of the vegetable insulating liquid. At the same time, the small amount of C_60_ nanoparticles lead to the poor inhibition of vegetable liquid cracking and the significantly increased dielectric loss of vegetable liquid. As the thermal aging of vegetable liquid progresses, when the traces of oxygen absorbed in the C_60_ nanoparticles is consumed, the C_60_ nanoparticles without oxygen demonstrate great resistance to the cracking of vegetable liquid, resulting in the low dielectric loss of the vegetable liquid during the later aging stage.

Figure 7 shows that electrical resistivity varies with different concentrations of C_60_ modified vegetable and mineral liquids. The electrical resistivity of the C_60_ modified vegetable liquid is higher than pure liquid in the early stage of aging. However, with the increased aging time, the addition of C_60_ nanoparticle reduced the value of electrical resistivity as compared to the pure liquid.

Triglyceride, a mixture of three fatty acid molecules and one glycerol molecule, is the main component of vegetable liquid, and fatty acid molecules consist of oleic acid, linoleic acid, α-linoleic acid, etc. Due to the sensitivity of the unsaturated double bonds of fatty acid molecules to oxygen at high temperature, the glycerol chains and the fatty acid molecules are easily oxidized and decomposed by oxygen at high temperature, thereby resulting in the generation of short-chain fatty acids, hydroxyl radicals, peroxides, ketones, aldehydes, and other substances. C_60_ nanoparticles as antioxidants are stronger than vitamin E composing of synthetic antioxidants, such as BHA and BHT. The addition of C_60_ nanoparticles to vegetable liquid can inhibit the action process of the hydroxyl radical and the hydroperoxide, thereby enhancing the oxidation resistance [38]. Unlike vegetable liquid, the mineral liquid is more difficult to shed hydrogen atoms from carbon chains due to the absence of labile double bonds. The oxidation induction period of mineral liquid is longer and the oxidation reaction is slightly intense as vegetable liquid.

### 3.5. Acid Values of Aged Nanofluid

Acid value is an important indicator to evaluate the oxidation degree of insulating liquid. Figure 8a shows that the acid value of nano vegetable liquid aging for 35 days was approximately 10 times larger than pure liquid at day 0. Figure 8a shows that pure liquid samples had lower acid value than C_60_ doped samples in the early accelerated thermal aging stage. This phenomenon indicates that modified liquids entered the development of the thermal acidification stage earlier than the pure sample. When the acid value of pure sample accelerated significantly, the trend of the curve evidently exceeded the modified liquid samples. Moreover, when additional C_60_ was added, the acid value was less after 20 days of aging. This result suggests that C_60_ nanoparticles inhibited the acidification process of vegetable liquid. Figure 8b shows that all mineral liquid samples possessed low acid level values during aging period, and the acid value of each sample did not exceed 0.05 mg (KOH)/g. The high concentration of C_60_ modified mineral liquid showed relatively low acid value.

### 3.6. Thermal Analysis of Aged Nanofluid

Quality variation is often observed when transferring materials in the heating process. The thermogravimetric analysis (TGA) tests the temperature control procedures and shows the relationship between the quality of the sample and test temperature. Differential thermal analysis (DTA), which reflects the endothermic and exothermic reactions of the test sample during the increase in temperature, is also measured. Owing to the deterioration of insulating liquid due to oxygen, 100 mg/L concentration of nanofluid was measured with nitrogen and air. The thermal analysis of C_60_ modified vegetable insulating liquid was carried out because the deterioration of insulating liquid was due to oxygen. The increase in temperature rate was 10 °C/min, and the flow rate of the atmosphere (nitrogen and air) was 50 mL/min. The C_60_ concentration of modified liquid was 100 mg/L, and each of the samples had been dried prior to the experiment. Thermal analysis provides two kinds of curves, namely, TGA and DTA.

Figure 9a shows the thermal analysis curves of the samples in the nitrogen atmosphere. The results show that both curves of nanofluid and pure liquid have similar changes. Each sample achieved maximum weight loss rate at nearly 415 °C, and maximum endothermic peak appeared at nearly 422 °C. All samples with or without C_60_ have identical thermal stability in the nitrogen atmosphere. Figure 9b shows the thermal analysis curves when the atmosphere is changed to air. The C_60_ modified liquid showed maximum weight loss rate earlier than the pure liquid, and maximum endothermic peak equally occurred in advance.

Figure 10 shows the thermal analysis curve of modified mineral liquid and pure liquid. In the presence of oxygen, the thermal analysis curves of nanofluid and pure liquid had a high degree of coincidence, with the maximum rate of weight loss which appeared near 220 °C and the maximum endothermic peak near 215 °C. This result shows that carbon has a slight effect on the thermal stability of mineral liquid.

### 3.7. Dissolved Gas Analysis of Aged Nanofluid

A few flammable gases, which were mostly dissolved in liquid, were generated when the insulation liquid was exposed to unusual thermal and electric fields in the transformer. The oil dissolved gas analysis (DGA) technology was used to effectively detect early failures within the transformer. This study investigated the gas production law of C_60_ modified insulating liquid, which accelerated thermal aging at 130 °C for 24 h in nitrogen.

Figure 11 shows the value of gas dissolved in pure vegetable liquid and nano vegetable liquid sample after thermal aging, respectively. The value of C_60_ modified vegetable liquid was lower than the pure sample, because the C_60_ nanoparticles inhibited the thermal decomposition of vegetable insulating liquid, thereby strengthening thermal stability.

Figure 12 shows the mechanism of anti-oxidation behavior of fullerene. C_60_ is expected to vanish radicals in the fluid under high temperatures by attaching radicals with double bonds on surface. In our previous study, it was confirmed that the vegetable fluid generated various radicals during thermal decomposition [39]. The fullerene with great ability of anti-oxidation can reduce the amounts of radicals, such as the C_3_H_5_• formed during the initial thermal decomposition of vegetable liquid and the H• generated for dissolved hydrogen in vegetable liquid. Thus, with the aid of fullerene, the thermal decomposition and hydrogen generation in vegetable liquid under high temperature can be inhibited greatly. The mechanism proposed corresponds to the dissolved gases vegetable liquid after thermal aging as shown in Figure 11.

Figure 13 shows the electron density distribution on C_60_ and the sketch figure for attracted electrons. The zones with blue colors stand for the positively charged area. The negative charge locates in the red area. The deeper color stands for greater electron density. It is observed that the positive charge locates in the core and on surfaces of the fullerene, and the negative charge locates around the positively charged area on surface of the fullerene. The electrons in the fluid under electrical stress are absorbed on surface of the fullerene by the positively charged area as shown in Figure 13. The reduced carrier concentration in the fluid leads to enhanced electrical performances including the breakdown performance, dissipation factor, and electrical resistivity. 

## 4. Conclusions

The present work focuses on the electrical properties and thermal stability of vegetable insulating liquid filled with C_60_ nanoparticles. The experimental and analytical results are concluded as follows:

The surface-modified C_60_ nanoparticles can be stably dispersed in insulating liquid. The electrical properties, such as dielectric loss factor, electrical resistivity, and breakdown voltage on nanofluids with different concentrations of C_60_ nanoparticles, were tested. The results showed that C_60_ nanoparticles enhanced the electrical properties with an optimum concentration of 100 mg/L, in which the dielectric loss factor decreased by 20.1%, electrical resistivity increased by 23.3%, and AC breakdown voltage increased by 8.6%. Meanwhile, the lightning impulse breakdown voltage increased by nearly 8%.

C_60_ molecules do not possess the capability to accelerate the thermal aging process of insulating liquid. Moreover, vegetable insulating liquid modified by C_60_ nanoparticles showed good thermal stability under the nitrogen atmosphere. Under the condition of oxygen, the hollow structure of the C_60_ nanoparticles provides a resident place for oxygen molecules, thereby resulting in the rapid deterioration of modified vegetable insulating liquid at the beginning of thermal aging. However, in the later stage of thermal aging, dielectric loss factor, volume resistivity, and the acid value level were improved compared to the pure liquid sample.

## Figures and Tables

**Figure 1 nanomaterials-09-00989-f001:**
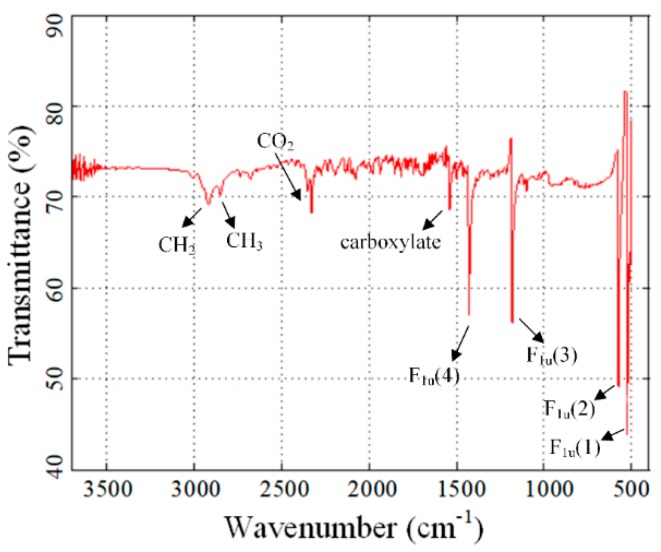
Infrared spectroscopy of the C_60_ nanoparticles.

**Figure 2 nanomaterials-09-00989-f002:**
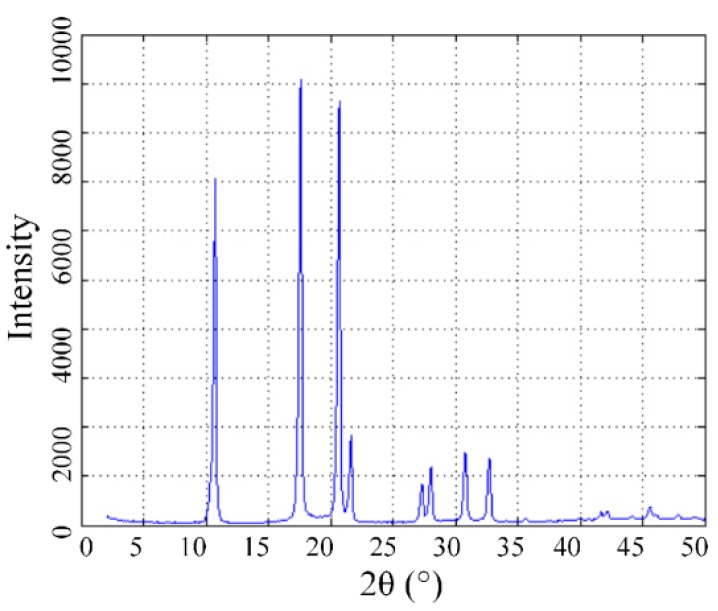
X-ray diffraction pattern of C_60_ after surface modification.

**Figure 3 nanomaterials-09-00989-f003:**
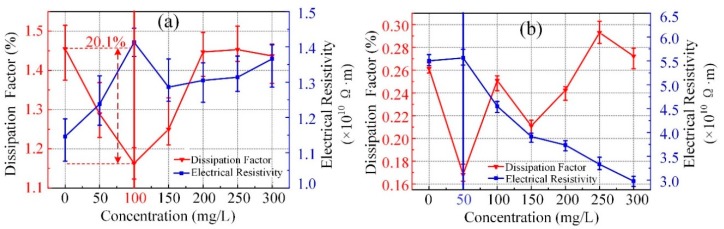
The variation of the dissipation factor and electrical resistivity with C_60_ concentrations of vegetable insulating liquid (**a**) and mineral liquid (**b**).

**Figure 4 nanomaterials-09-00989-f004:**
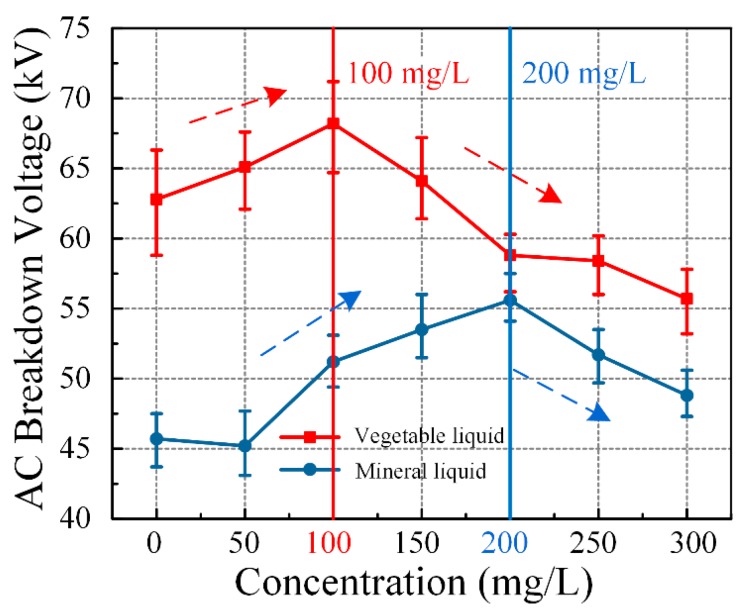
The variation of AC breakdown voltage with C_60_ concentrations of vegetable insulating liquid and mineral liquid.

**Figure 5 nanomaterials-09-00989-f005:**
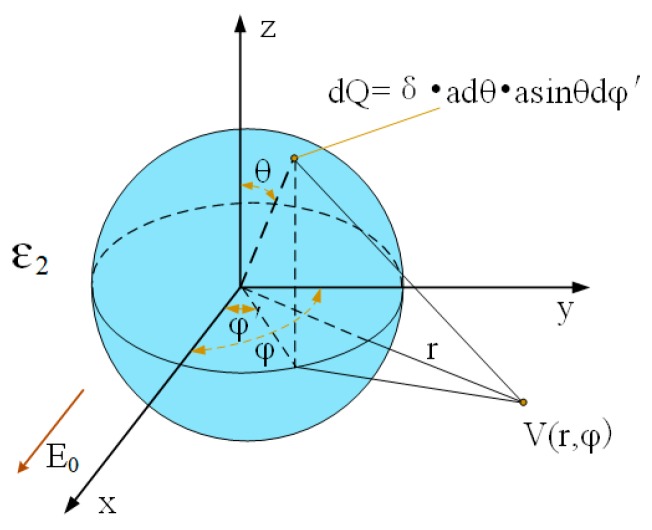
Surface polarization of charged nanoparticles in a continuous dielectric field.

**Figure 6 nanomaterials-09-00989-f006:**
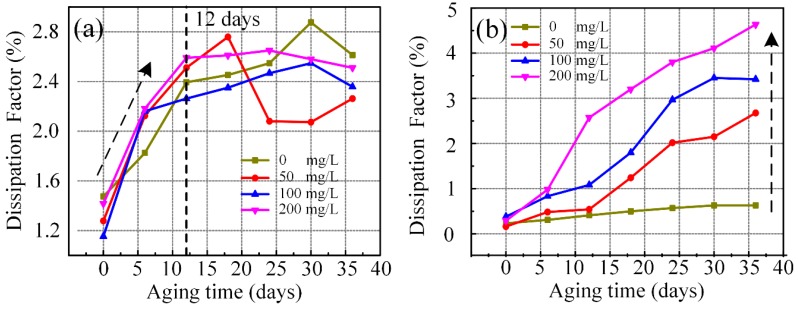
The variation of dissipation factor with C_60_ concentration of aging vegetable insulating liquid (**a**) and aging mineral liquid (**b**).

**Figure 7 nanomaterials-09-00989-f007:**
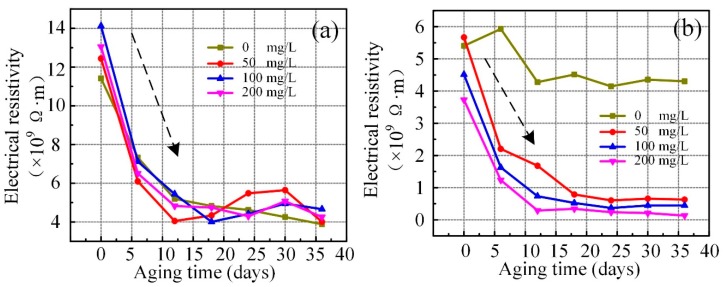
The variation of electrical resistivity with C_60_ concentration of aging vegetable insulating liquid (**a**) and aging mineral liquid (**b**).

**Figure 8 nanomaterials-09-00989-f008:**
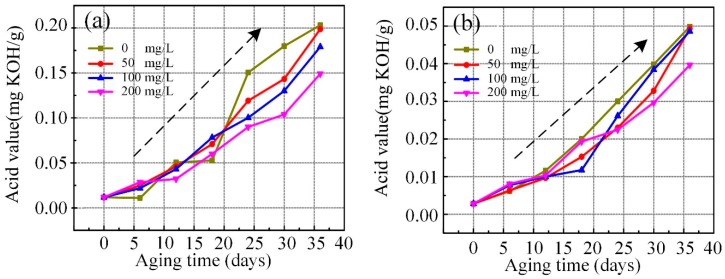
The variation of acid with C_60_ concentration of aging vegetable insulating liquid (**a**) and aging mineral liquid (**b**).

**Figure 9 nanomaterials-09-00989-f009:**
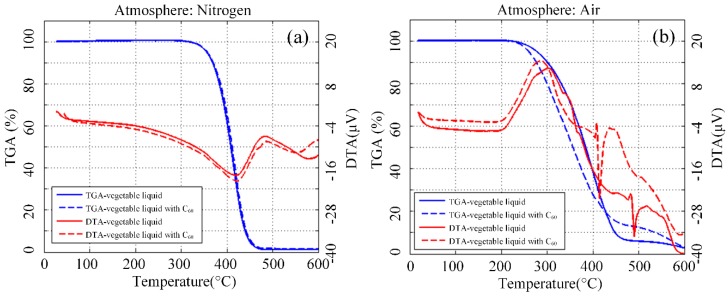
TGA and DTA curve of C_60_ modified vegetable insulating liquid and its pure sample in nitrogen (**a**) and air (**b**) atmosphere.

**Figure 10 nanomaterials-09-00989-f010:**
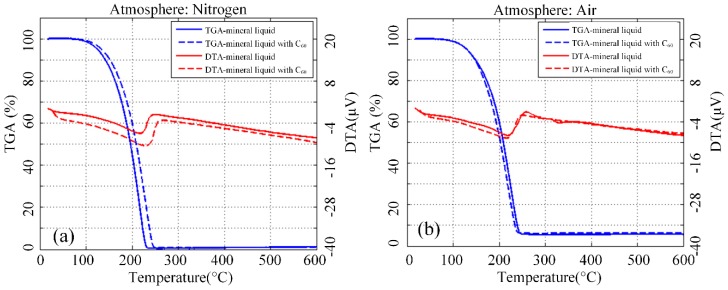
TGA and DTA curve of C_60_ modified mineral insulating liquid and pure sample nitrogen (**a**) and air (**b**) atmosphere.

**Figure 11 nanomaterials-09-00989-f011:**
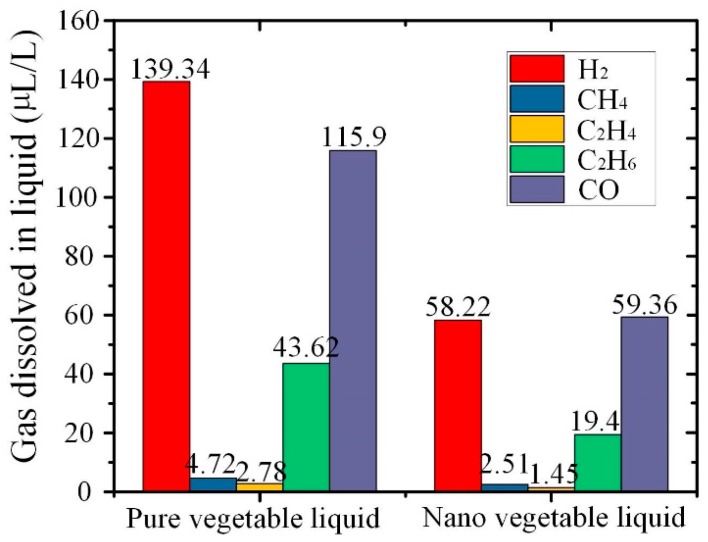
Dissolved gases of vegetable insulating liquid after thermal aging.

**Figure 12 nanomaterials-09-00989-f012:**
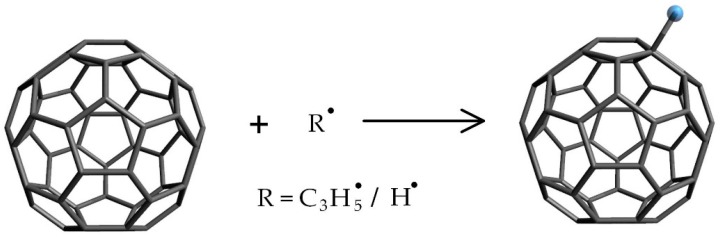
The mechanism of the anti-oxidation behavior of fullerene.

**Figure 13 nanomaterials-09-00989-f013:**
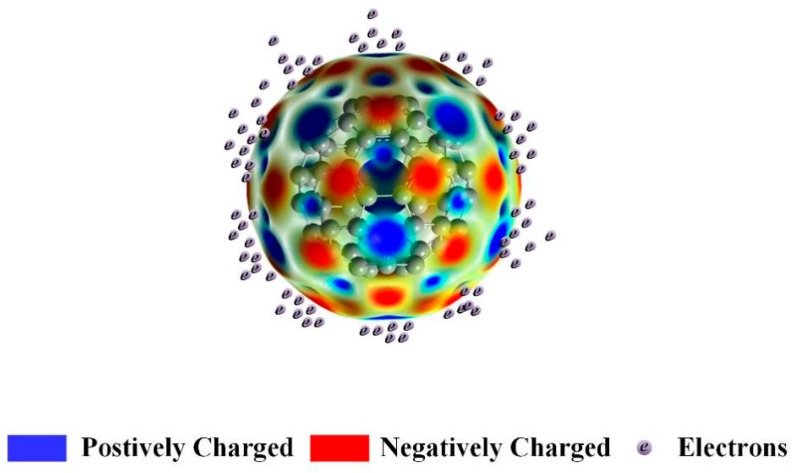
The calculation of electron density distribution on fullerene and the sketch figure for attracted electrons.

**Table 1 nanomaterials-09-00989-t001:** Basic physicochemical properties of insulating liquid.

Parameters	Unit Symbol	Typical Value	
		Vegetable Liquid	Mineral Liquid
Kinematic Viscosity at 40 °C	mm^2^·s^−1^	41.0	10.0
Density at 20 °C	kg·m^−3^	0.90	0.83–0.89
Flash Point	°C	320	≥135
Pour Point	°C	−18	−22
Acid Value	mgKOH·g^−1^	≤0.03	≤0.01

**Table 2 nanomaterials-09-00989-t002:** Lightning impulse breakdown voltage of insulating liquid.

Liquid Types	Breakdown Voltage (kV)		Breakdown Time (μs)	
	Positive	Negative	Positive	Negative
Vegetable Liquid	78.2	83.7	10.3	11.9
Nano Vegetable Liquid	83.9	89.9	10.9	12.3
Mineral Liquid	60.8	103.3	8.7	11.1
Nano Mineral Liquid	66.9	111.2	9.2	11.9

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
