# Peer review of "Significantly Enhanced Electrical Performances of Eco-Friendly Dielectric Liquids for Harsh Conditions with Fullerene"

_nanomaterials, 2019, doi:10.3390/nano9070989_

Round 1
Reviewer 1 Report
In presented manuscript authors deal with electrical properties of nanofluids. In my opinion, the topic of the paper is important and definitely worth of investigation. However, at the moment paper has some significant issues that make it unpublishable and major revision should be required.
There are two things that must be discussed by authors before evaluation of the paper:
1. The measuring procedure description must be extend. The detailed discussion of the uncertainty, its evaluation, must be presented. There is no information about it, and later on the graphs some uncertainty bars are presented...
2. What was (and how it was determined) real concentration of particles in nanofluids. In text authors said it was for example 300 ppm (parts per million), while on the graphs we can see 300 micro grams per litre.
Without presenting those information it is impossible to evaluate scientific value of this research.
During revision authors should also improve some other parts of manuscript (minor changes), like:
1. In the introduction section authors discuss the breakdown voltage of nanofluids, while they omitted recent state-of-art review paper on this issue presented by Fal et al.: "Nanofluids in the Service of High Voltage Transformers: Breakdown Properties of Transformer Oils with Nanoparticles, a Review" Energies, DOI: https://doi.org/10.3390/en11112942 I think this paper should be presented in the Introduction section.
2. In section 2.1 the ultrasound energy should be presented. Did authors control temperature of the sample during sonication?
3. A coma "," or dot "." should be putted after each equation. For example eq. (1) is just part of sentence, so the coma should be placed after it.
4. The source of the data presented in Table 1 must be presented. If those values were measured by the authors - the measuring procedure and uncertainty should be described.
5. I suggest to use well established subscripts to describe nanofluids (nf), nanoparticles (np) and base fluid (bf) instead of "1" and "2" in equations 2-4.
In conclusion, paper presents some significant problems that must be resolved before its evaluation, but is interesting enough to be revised.
Reviewer 2 Report
In the last paragraph of Introduction there would be good to say, what is this vegetable insulating liquid, the content and proportions...
In the preparation section there is lack of preparation procedure of the liquid and the mineral liquid (was it bought or prepared on site)
r.173: 1.3x10^10
r.280-287 please clarify and check the English
Fig12, 13 were calculated by Gaussian? I have doubts about ploted elctrons - the density is nonhomogenious...
Reviewer 3 Report
This is an interesting work where C60 molecules do not possess the capability to accelerate the thermal aging process of insulating liquid but does improve the electrical properties. Moreover, vegetable insulating liquid modified by C60 nanoparticles shows good thermal stability under the nitrogen atmosphere. Under the condition of oxygen, the hollow structure of the C60 nanoparticles provides a resident place for oxygen molecules, thereby resulting in the rapid deterioration of modified vegetable insulating liquid at the beginning of thermal aging. However, in the later stage of thermal aging, dielectric loss factor, volume resistivity, and acid value level were improved as compared to those of the pure liquid sample.
Authors can reduce the verbose description in the introduction as cited references are good enough; brevity is needed.
Fig. 1 appears very poor with limited information; can a better picture with description of bands be included?
This reviewer could not help but wonder what happens with other carbon materials (for comparison) such as graphitic carbon nitrides or graphene (reduced and graphene oxide)?
Round 2
Reviewer 1 Report
Authors make the great improvement of the manuscript. It may be accepted as it is now.
Author Response
We would like to extend our sincere appreciation to the reviewer for the kind advice.
Reviewer 3 Report
Authors have addressed my concerns but the following newly added statement 'Highlighted' need to be corrected, Lines 286-290: "On the other hand, because of unsaturated double bonds in the molecular chains of oleic acid, linoleic acid, and α-linoleic acid, the fatty acid molecules and glycerol chains will produce decomposition free from the effect of oxygen and temperature, thereby resulting in the generation of chain-breaking fatty acids, hydroxyl radicals, peroxides, ketones, aldehydes, and other substances.
